# MetaBox: A Benchmark Platform for Meta-Black-Box Optimization with Reinforcement Learning

**Zeyuan Ma**[1], **Hongshu Guo**[1], **Jiacheng Chen**[1], **Zhenrui Li**[1], **Guojun Peng**[1],
**Yue-Jiao Gong**[1,*], **Yining Ma**[2], **Zhiguang Cao**[3]
[1]South China University of Technology
[2]National University of Singapore
[3]Singapore Management University
{scut.crazynicolas, guohongshu369, jackchan9345}@gmail.com,
zhenrui.li@outlook.com, {pgj20010419, gongyuejiao}@gmail.com,
yiningma@u.nus.edu, zhiguangcao@outlook.com

## Abstract

Recently, Meta-Black-Box Optimization with Reinforcement Learning (MetaBBO-RL) has showcased the power of leveraging RL at the meta-level to mitigate manual fine-tuning of low-level black-box optimizers. However, this field is hindered by the lack of a unified benchmark. To fill this gap, we introduce MetaBox, the first benchmark platform expressly tailored for developing and evaluating MetaBBO-RL methods. MetaBox offers a flexible algorithmic template that allows users to effortlessly implement their unique designs within the platform. Moreover, it provides a broad spectrum of over 300 problem instances, collected from synthetic to realistic scenarios, and an extensive library of 19 baseline methods, including both traditional black-box optimizers and recent MetaBBO-RL methods. Besides, MetaBox introduces three standardized performance metrics, enabling a more thorough assessment of the methods. In a bid to illustrate the utility of MetaBox for facilitating rigorous evaluation and in-depth analysis, we carry out a wide-ranging benchmarking study on existing MetaBBO-RL methods. Our MetaBox is open-source and accessible at: https://github.com/GMC-DRL/MetaBox.

## 1 Introduction

Black Box Optimization (BBO) is a class of optimization problems featured by its objective function that is either unknown or too intricate to be mathematically formulated. It has a broad range of applications such as hyper-parameter tuning [1], neural architecture searching [2], and protein-docking [3]. Due to the *black-box* nature, the optimizer has no access to the mathematical expression, gradients, or any other structural information related to the problem. Instead, the interaction with the black-box problem is primarily realized through querying inputs (i.e., a solution) and observing outputs (i.e., its objective value).

Traditional solvers for BBO problems include population-based optimizers such as genetic algorithms [4], evolutionary strategies [5–7], particle swarm optimization [8, 9], and differential evolution [10–13]. The Bayesian Optimization (BO) [14, 15] is also commonly used. However, with limited knowledge of the problem, these optimizers lean on carefully hand-crafted designs to strike a balance between exploration and exploitation when seeking the optimal solution.

To eliminate the burdensome task of manual fine-tuning, recent research has proposed the concept of **Meta-Black-Box Optimization (MetaBBO)**, which aims to refine the black-box optimizers by

---

*Yue-Jiao Gong is the corresponding author.

37th Conference on Neural Information Processing Systems (NeurIPS 2023) Track on Datasets and Benchmarks.

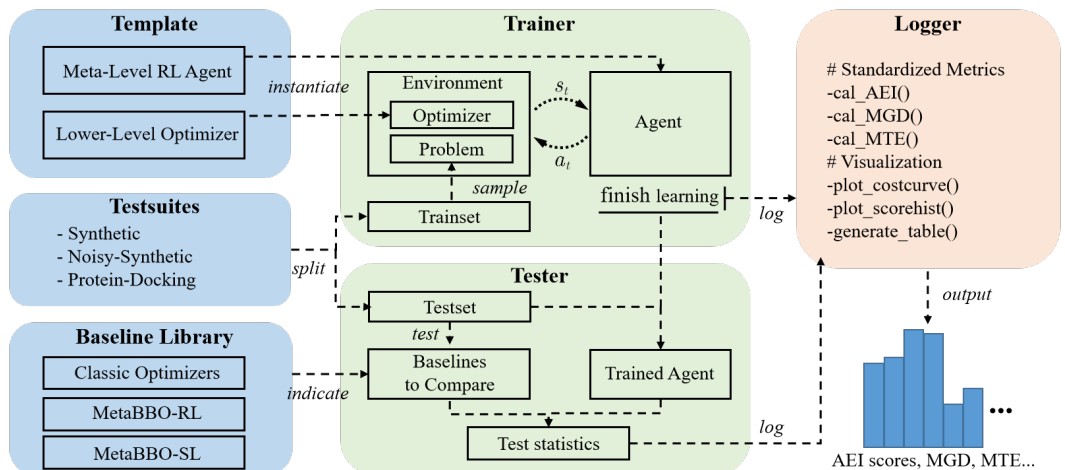

Figure 1: Blueprint of our MetaBox platform. MetaBox offers template scripts for quick-start and, once completed by the user, it carries out an automatic Train-Test-Log process on the testsuites for comparison. Results such as performance scores, optimization curves, and comparative tables are automatically generated and made available for review.

identifying optimal configurations or parameters that boost the overall performance across various problem instances within a given problem domain. This leads to a bi-level optimization framework, where the meta-level enhances the performance of low-level black-box optimizers. The meta-level approaches include supervised learning (MetaBBO-SL) [16–18], reinforcement learning (MetaBBO-RL) [19–25], and self-referential search (MetaBBO-SR) [26, 27]. Among them, MetaBBO-RL models the optimizer fine-tuning as a Markov Decision Process (MDP) and learns an agent to automatically make decisions about the algorithmic configurations. By automating the tuning process, MetaBBO-RL not only significantly reduces the time and effort needed for customizing algorithms to specific unseen problems, but also potentially enhances the overall optimization performance.

Despite the success, MetaBBO-RL calls for a unified benchmark platform. Upon examining the recently proposed MetaBBO-RL approaches [19–25], we found that while several approaches claim state-of-the-art performance, they lack a comprehensive benchmark and comparison using a standardized, unified testbed. As a result, identifying which approach truly excels under specific conditions poses a significant challenge, thereby hindering further progress and advancement in this field.

To bridge the gap, we introduce MetaBox, the first benchmark platform for MetaBBO-RL. The blueprint of MetaBox is illustrated in Figure 1, aiming to offer researchers a convenient means to develop and evaluate the MetaBBO-RL approaches. Specifically, we have made the following efforts:

1. **To simplify the development of MetaBBO-RL and ensure an automated workflow:** 1) We introduce a *MetaBBO-RL Template*, depicted at the top left of Figure 1. It comprises two main components: the meta-level RL agent and the low-level optimizer, where we provided several implemented works (e.g., [19, 21, 24, 25]) following a unified interface protocol. 2) We automate the *Train-Test-Log* procedure in MetaBox, shown in the middle of Figure 1. The users simply need to complete their *MetaBBO-RL template*, specify the target problem set, and select several baselines for comparison. Initiating the entire automated process is as straightforward as executing the *run_experiment()* command. This provides users with considerable flexibility in implementing different types of internal logic for their MetaBBO-RL algorithms while benefiting from the automated workflow.

2. **To facilitate broad and standardized comparison studies:** 1) We integrate a large-scale *MetaBox testsuite* consisting of over 300 benchmark problems with diverse landscape characteristics (such as single-/multi- modal, non-ill/ill- conditioned, strong/weak global structured, and noiseless/noisy). Showing in the left centre of Figure 1, the *MetaBox testsuite* inherits problem definitions from the well-known COCO [28] platform and the Protein-Docking benchmark (version 4.0) [29], with several modifications to adapt to the MetaBBO paradigms. 2) We develop a *Baseline Library*, located at the bottom left in

Figure 1. The library currently encompasses a wide range of classic optimizers [6, 8–14, 30, 31] and up-to-date MetaBBO-RL approaches [19–25]. We additionally integrate a MetaBBO-SL approach [16] to provide an extended comparison. Notably, all of the baselines are implemented by our *MetaBBO-RL Template*. This ensures consistency and allows for a fair and standardized comparison among the different approaches.

3. **To comprehensively evaluate the effectiveness of MetaBBO-RL approaches:** 1) We propose three *Standardized Metrics* to evaluate both optimization performance and learning effectiveness of a MetaBBO-RL approach, including a novel Aggregated Evaluation Indicator (AEI) that offers a holistic view of the optimization performance, a Meta Generalization Decay (MGD) metric that measures the generalization of a learned approach across different problems, and a Meta Transfer Efficiency (MTE) metric that quantifies the transfer learning ability. 2) We conduct a tutorial large-scale comparison study using *Baseline Library*, evaluate them on *MetaBox testsuite* by the proposed *Standardized Metrics*. Several key findings reveal that the pursuit of state-of-the-art performance continues to present challenges for current MetaBBO-RL approaches. Nonetheless, our MetaBox platform provides valuable insights and opportunities for researchers to refine and improve their algorithms.

To summarize, MetaBox provides the first benchmark platform for the MetaBBO-RL community (**novel**). It is fully open-sourced, offering template scripts that facilitate convenient development, training, and evaluation of MetaBBO-RL algorithms (**automatic**). Furthermore, MetaBox provides diverse benchmark problems and an extensive collection of integrated baseline algorithms, which will continue to expand through regular maintenance and updates (**extendable**).

## 2 Background and Related Work

As shown in 2, MetaBBO methods operate within a bi-level optimization framework designed to automate the fine-tuning process for a given BBO optimizer. Distinguishing themselves from conventional BBO techniques, MetaBBO methods introduce a novel meta-level as an automatic decision process. The purpose is to alleviate the need for labor-intensive manual fine-tuning of low-level BBO optimizers. Typically, they require the ability to generalize behaviors to address previously unseen problems through extensive training in a given problem distribution. Concretely: **1) At the meta level**, the meta optimizer (e.g., an RL agent) dynamically configures the low-level optimizer based on the current optimization status at that particular time step. Then, the meta optimizer evaluates the performance of the low-level optimizer over the subsequent optimization steps, referred to as meta performance. The meta optimizer leverages this observed meta performance to refine its decision-making process, training itself through the maximization of accumulated meta performance, thereby advancing its meta objective. **2) At the lower level**, the BBO optimizer receives a designated algorithmic configuration from the meta optimizer. With this configuration in hand, the low-level optimizer embarks on the task of optimizing the target objective. It observes the changes in the objective values across consecutive optimization steps and transmits this information back to the meta optimizer, thereby contributing to the meta performance signal.

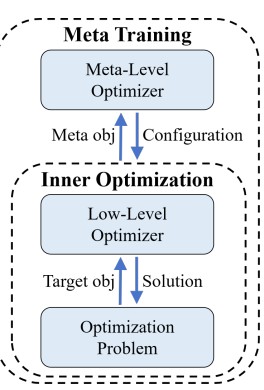

Figure 2: Illustration of the bi-level optimization procedure of MetaBBO.

**MetaBBO-RL.** It leverages reinforcement learning at the meta-level to configure the low-level black-box optimizer for boosting meta-performance [26] over a problem distribution. It involves three components: an RL agent (policy) $\Pi_\theta$, a backbone black-box optimizer $\Lambda$, and a dataset $\mathbb{D}$ following certain problem distribution. The environment $Env$ in MetaBBO-RL is formed by coupling the optimizer with a problem from the dataset, i.e., $Env := \{\Lambda, f = sample(\mathbb{D})\}$, where $f$ represents the sampled problem. Unlike traditional RL tasks [32, 33], the environments in MetaBBO-RL not only include the problem of optimizing but also include the low-level optimizer itself. At each step $t$, the agent queries the optimization status $s_t$, which includes the current solutions produced by $\Lambda$ and their evaluated values by $f$. The policy $\Pi_\theta$ takes $s_t$ as input and suggests an action $a_t = \Pi_\theta(s_t)$ to configure the optimizer $\Lambda$. Note that the action space of different MetaBBO algorithms can vary largely, as it could involve determining solution update strategy [19, 20, 23], generating hyper-parameter [21, 24, 25], or both of the above two [22]. Subsequently, $Env$ executes $a_t$ to obtain the

Table 1: Comparison to BBO benchmarks. We report *#Problem*: the number of problems (*#synthetic + #realistic*); *#Baseline*: the number of baselines; *Template*: Template coding support; *Automation*: automated train/test workflow support; *Customization*: configurable settings; *Visualization*: visualization tools support; and *RLSupport*: Gym-style [33] RL benchmark.

| | *#Problem* | *#Baseline* | *Template* | *Automation* | *Customization* | *Visualization* | *RLSupport* |
|---|---|---|---|---|---|---|---|
| COCO [28] | 54+0 | 2 | ✓ | ✓ | ✗ | ✓ | ✗ |
| CEC [34, 36, 37] | 28+0 | 0 | ✗ | ✗ | ✗ | ✗ | ✗ |
| IOHprofiler [38, 39] | 24+0 | 0 | ✓ | ✗ | ✓ | ✓ | ✗ |
| Bayesmark [40, 41] | 0+228 | 10 | ✓ | ✓ | ✗ | ✗ | ✗ |
| Zigzag [42, 43] | 4+0 | 0 | ✗ | ✗ | ✓ | ✗ | ✗ |
| MetaBox | 54+280 | 19 | ✓ | ✓ | ✓ | ✓ | ✓ |

next optimization status $s_{t+1}$ and the reward $r_t$ that measures the meta-performance improvement of $\Lambda$ on problem $f$. The meta-objective of MetaBBO-RL is to learn a policy $\Pi_\theta$ that maximizes the expectation of the accumulated meta-performance improvement $r_t$ over the problem distribution $\mathbb{D}$, $\mathbb{E}_{f \sim \mathbb{D}, \Pi_\theta}[\sum_{t=0}^{T} r_t]$, where $T$ denotes the predefined evaluation budget, typically referred to the *maxFEs* parameter as in the existing BBO benchmarks [28, 34].

**Other MetaBBO methods.** Apart from MetaBBO-RL, two other paradigms include MetaBBO-SL [16–18] and MetaBBO-SR [26, 27], which all belong to the MetaBBO community but leverage different approaches at the meta-level. Existing MetaBBO-SL methods consider a recurrent neural network (RNN) that helps determine the next solutions at each step [16–18]. However, they often encounter a dilemma in setting the horizon of RNN training: a longer length improves the difficulty of training, whereas a shorter length requires breaking the entire optimization process into small pieces, often resulting in unsatisfactory results. MetaBBO-SR utilizes black-box optimizers (such as an evolution strategy [35]) at both the meta and low levels to enhance the overall optimization performance. Since the black-box optimizers themselves can be computationally expensive, the MetaBBO-SR approaches may suffer from limited efficiency due to their inherently nested structure.

**Related BBO benchmarks.** Existing benchmarks for MetaBBO are currently absent. However, there are several related BBO benchmark platforms, including COCO [28] and IOHprofiler [38, 39] for continuous optimization, ACLib [44] for algorithm configuration, Olympus [45] for planning tasks, and Bayesmark [40] for hyper-parameter tuning. These platforms provide testsuites, logging tools and some baselines for users to compare or refer to. In addition to these platforms, some BBO competitions provide a series of problems to evaluate specific aspects of algorithms, such as the GECCO BBOB workshop series (based on COCO [28]), the NeurIPS BBO challenge [41] (based on Bayesmark [40]), the IEEE CEC competition series [36], and the Zigzag BBO [42, 43] that consists of highly challenging problems. In contrast, our proposed MetaBox introduces a novel benchmark platform specifically tailored for MetaBBO-RL. It provides an algorithm development template, an automated execution philosophy, a wide range of testsuites, an extensive collection of baselines, specific MetaBBO performance metrics, and powerful visualization tools. In Table 1, we compare MetaBox and other related benchmarks to showcase the novelty of this work.

## 3    MetaBox: Design and Resources

### 3.1    Template coding and workflow automation

The core structure of MetaBox is presented on the left of Figure 3, shown in the form of a UML class diagram [46]. Drawing from the recent MetaBBO-RL, we abstract the *MetaBBO-RL Template* into two classes: the reinforcement *Agent* and the backbone *Optimizer* (both marked in pale orange). To develop or integrate a new MetaBBO-RL approach, users are required to specify their settings in the attributes $Agent.config$ and $Optimizer.config$, and implement the following interfaces: 1) $Agent.train\_episode(env)$ for the training procedure, 2) $Agent.rollout\_episode(env)$ for the rollout procedure, and 3) $Optimizer.update(action, problem)$ for the optimization procedure. Once these interfaces are set, the implemented templates seamlessly integrate with other components, requiring no extra adjustments. Except for the *Agent* and *Optimizer* classes, the other classes are hidden from users, simplifying coding tasks and ensuring code consistency.

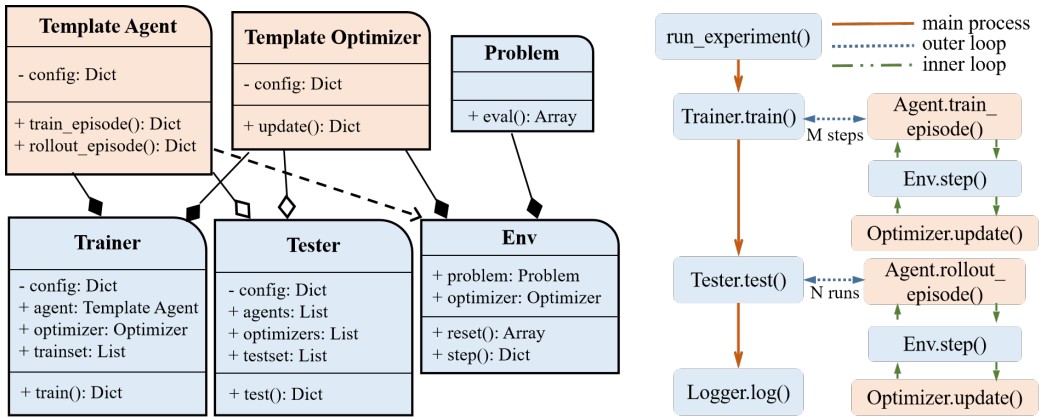

Figure 3: The core structure and workflow of MetaBox. **Left:** UML Class diagram of MetaBox. Users can inherit from MetaBBO-RL templates highlighted in orange to enable polymorphism. **Right:** The automated Train-Test-Log workflow. The Agent directs the low-level (inner loop) optimization and trains itself on meta-level (outer loop), followed by testing and post-processing.

Given an implemented approach based on *MetaBBO-RL Template*, the right side of Figure 3 illustrates the entirely automated *Train-Test-Log* workflow in MetaBox. The process initiates when a user launches the function $run\_experiment()$. Then, function $Trainer.train()$ executes an outer-loop for meta-level iterations, where a runtime RL environment ($env$) is instantiated by pairing a problem from the trainset with an optimizer instance. The $Agent.train\_episode(env)$ function is then called to start an inner-loop for the BBO iterations, during which the optimizer in $env$ follows the $action$ informed by $Agent$ and applies its $update(action, problem)$ function to optimize the target BBO problem until the predefined function evaluation budget is exhausted. The meta-iteration ends when the $Agent$ has trained its policy model for $M$ steps. Next, the $Tester.test()$ function is called to evaluate the learned policy on the testset through $N$ independent runs, recording the statistic results. Finally, the $Logger.log()$ function restores all saved training and testing results. These basic results are further post-processed into comprehensive metrics for in-depth analysis. The key to this automation resides in the universal internal-call bridging the *MetaBBO-RL Template* and the *Train-Test-Log* workflow, thereby obviating the need for users to grapple with complex coding tasks.

## 3.2 Testsuites

The testsuites integrated into MetaBox are briefly described as follows (more details in Appendix A):

**Synthetic.** It consists of 24 functions in five groups: separable, moderate, ill-conditioned, multi-modal with global structure, and multi-modal with weakly structured. They are collected from the *coco:bbob* function set in the renowned COCO platform [28].

**Noisy-Synthetic.** It consists of 30 noisy functions from the *coco:bbob-noisy* function set [28], by extending the Synthetic set with different noise models and levels. The noises come from three different models: Gaussian noise, Uniform noise and Cauchy noise [47].

**Protein-Docking.** It is extracted from the Protein-Docking benchmark [29], with 280 instances of different protein-protein complexes. These problems are characterized by rugged objective landscapes and are computationally expensive to evaluate.

Besides, we make the following efforts to adapt the above testsuites, while enhancing usability and flexibility: 1) We treat each testsuite as a dataset and split it into training and testing sets according to a particular proportion, referred to as *difficulty*. This proportion determines the level of difficulty in generalizing or transferring the learned knowledge to unseen instances during testing. We provide two modes for controlling this aspect: *easy* mode, allocating 75% of the selected instances for training, and *difficult* mode, designating 25% of the selected instances for training. 2) Instead of providing predefined problem dimension candidates, such as {2,3,5,10,20,40} in COCO [28] and {10,30} in CEC2021 [34], we introduce an additional control parameter *problem-dim* to support customized problem dimension. 3) We also recognize that there is a potential user group (e.g., those in [16–18])

who require access to gradients of the objective during the training of their approaches[2]. To cater to this need, we provide a PyTorch function interface with tensor calculation support, enabling users to incorporate back-propagation in their methods.

## 3.3 Baseline library

MetaBox leverages a total of 19 baselines, categorized into three types: 1) **Classic Optimizer:** Random Search (RS), Bayesian Optimization (BO) [14], Covariance Matrix Adaptation Evolution Strategy (CMA-ES) [6], Differential Evolution (DE) [10] and its self-adaptive variants JDE21 [12], MadDE [11], NL-SHADE-LBC [13], Particle Swarm Optimization (PSO) [8] and its self-adaptive variants GLPSO [9], sDMSPSO [30], SAHLPSO [31]. 2) **MetaBBO-RL:** the Q-learning [48–50] styled approaches DEDDQN [19], DEDQN [20], RLHPSDE [22], QLPSO [23], and the Policy Gradient [51, 52] styled methdos LDE [21], RLPSO [24], RLEPSO [25]. 3) **MetaBBO-SL:** RNN-OI [16].

The majority of the classic baselines are reproduced by referring to the originally released codes (when available) or the respective original papers. We meticulously filled the *Optimizer* template (see Section 3.1) with their specific internal logic. However, there are a few exceptions: CMA-ES, DE and PSO are implemented by calling APIs of DEAP [53] (an evolutionary computation framework with LGPL-3.0 license), and BO is implemented by the Scikit-Optimizer [54] (a BO solver set with BSD-3-Clause license). This showcases the compatibility of MetaBox with existing open-sourced optimization codebases. For MetaBBO-RL, we first encapsulate the internal logic of the RL agent into the *Agent* template (see Section 3.1), and then encapsulate the internal logic of the backbone optimizer into the *Optimizer* template. This showcases the ease of integrating various MetaBBO-RL methods into MetaBox. In addition, we provide an example of a MetaBBO-SL approach, RNN-OI, to showcase that MetaBox is compatible with other MetaBBO methods.

## 3.4 Performance metrics

In MetaBox, we implement three standardized metrics to evaluate the performance of MetaBBO-RL approaches from aspects of both the BBO performance and the training efficacy.

**AEI.** The MetaBBO approaches can be evaluated using traditional BBO performance metrics, including the best objective value, the budget to achieve a predefined accuracy (convergence rate), and the runtime complexity [28, 36, 39]. In addition to analyzing them separately, we propose the Aggregated Evaluation Indicator (AEI), a unified scoring system to provide users with a comprehensive assessment. Aggregating these metrics together is challenging due to the significant variation in values across different metrics and problems. To address this, the AEI normalizes and combines the three metrics in the following way. We test a MetaBBO-RL approach on $K$ problem instances for $N$ repeated runs and then record the basic metrics (usually pre-processed by a min-max conversion for consistency towards AEI maximization, refer to Appendix B for details): best objective value $v_{obj}^{k,n}$, consumed function evaluation times $v_{fes}^{k,n}$, and runtime complexity $v_{com}^{k,n}$. To make the values more distinguishable and manageable, they have been first subjected to a logarithmic transformation:

$$v_*^{k,n} = \log(v_*^{k,n}). \tag{1}$$

Then, Z-score normalization is applied:

$$Z_*^k = \frac{1}{N} \sum_{n=1}^{N} \frac{v_*^{k,n} - \mu_*}{\sigma_*}, \tag{2}$$

where $\mu_*$ and $\sigma_*$ are calculated by using RS as a baseline. Finally, the AEI is calculated by:

$$AEI = \frac{1}{K} \sum_{k=1}^{K} e^{Z_{obj}^k + Z_{com}^k + Z_{res}^k}, \tag{3}$$

where Z-scores are first aggregated, then subjected to an inverse logarithmic transformation, and subsequently averaged across the test problem instances. A higher AEI indicates better performance of the corresponding MetaBBO-RL approach.

---

[2]Important to note that the gradients of the objective during testing are not allowed in the context of BBO.

**MGD.** We then introduce the Meta Generalization Decay (MGD) metric, so as to assess the generalization performance of MetaBBO-RL for unseen tasks. Given a model that has been trained on a problem set $B$ and its AEI on the corresponding testset as $AEI_B$, we train another model on a problem set $A$ and record its AEI on the testset $B$ as $AEI_A$. The $MGD(A, B)$ is computed by:

$$MGD(A, B) = 100 \times (1 - \frac{AEI_A}{AEI_B})\%, \tag{4}$$

where a smaller $MGD(A, B)$ indicates that the approach generalizes well from $A$ to $B$. Note that MGD has neither symmetry nor transitivity properties.

**MTE.** When zero-shot generalization is unachievable due to significant task disparity, the Meta Transfer Efficiency (MTE) metric is proposed to evaluate the transfer learning capacity of a MetaBBO-RL approach. We begin by maintaining checkpoints of a MetaBBO-RL approach trained on a problem set $B$. We locate the checkpoint with the highest cumulative return, recording its index as $T_{\text{scratch}}$. Next, we pre-train a checkpoint of the MetaBBO-RL approach on another problem set $A$, then load this checkpoint back and continue the training on the problem set $B$ until it reaches the same best-accumulated return, recording the current index as $T_{\text{finetune}}$. The $MTE(A, B)$ is calculated by:

$$MTE(A, B) = 100 \times (1 - \frac{T_{\text{finetune}}}{T_{\text{scratch}}})\%, \tag{5}$$

where a larger $MTE(A, B)$ indicates that the knowledge learned in $A$ can be easily transferred to solve $B$, while an MTE value less than or equal to zero indicates potential negative transfer issues. Similar to MGD, the MTE has neither symmetry nor transitivity properties.

## 4 Benchmarking Study

MetaBox serves as a valuable tool for conducting experimental studies, allowing researchers to 1) benchmark specific groups of MetaBBO algorithms, 2) tune their algorithms flexibly, and 3) perform in-depth analysis on various aspects including generalization and transfer learning abilities. In this section, we provide several examples to illustrate the use of MetaBox in experimental studies.

### 4.1 Experimental setup

To initiate the fully automated *Train-Test-Log* process of MetaBox, users need to follow two steps: 1) indicate environment parameters, including *problem-type*, *problem-dim* and *difficulty*, to inform MetaBox about the problem set, its dimension and the train-test split used in the comparison study; and 2) specify the maximum number of training steps ($M$), the number of independent test runs ($N$), and the reserved function evaluation times (*maxFEs*) for solving a problem instance. Once these two steps are completed, users can trigger the *run_experiment()* to enjoy the full automation process. In the exemplary study below, $M$, $N$ and *maxFEs* are set to $1.5 \times 10^6$, 51, and $2 \times 10^4$, respectively, unless specified otherwise. All results presented are obtained using a machine of Intel i9-10980XE CPU with 32GB RAM. Note that MetaBox is also compatible with other platforms. We provide default control parameters for each baseline, which are listed in Appendix C.

### 4.2 Comparison of different baseline (Meta)BBO methods

We train all MetaBBO baselines (7 MetaBBO-RL and 1 MetaBBO-SL) in the *Baseline Library* on Synthetic and Protein-Docking testsuites, using both *easy* and *difficult* train-test split. We then test these baselines against the classic black-box optimizers available in our library and calculate the AEI of each algorithm. The results are depicted in Figure 4. On the Synthetic testsuites, two MetaBBO-RL methods, namely LDE and RLEPSO, show competitive performance against classic baselines. Specifically, LDE outperforms JDE21, the winner of IEEE CEC 2021 BBO Competitions [34], on the Synthetic-difficult testsuites, while RLEPSO outperforms both JDE21 and NL-SHADE-LBC, the winner of IEEE CEC 2022 BBO Competitions [55], on the Synthetic-easy testsuites. However, for this particular problem set, CMA-ES remains a strong baseline. As we shift to more realistic testsuites (Protein-docking of 280 instances), most baselines show varying degrees of performance drop. Notably, CMA-ES is inferior to a number of algorithms such as DEDQN, LDE, RLEPSO, and GLPSO. In general, we observe that MetaBBO-RL methods are more robust than classic hand-crafted

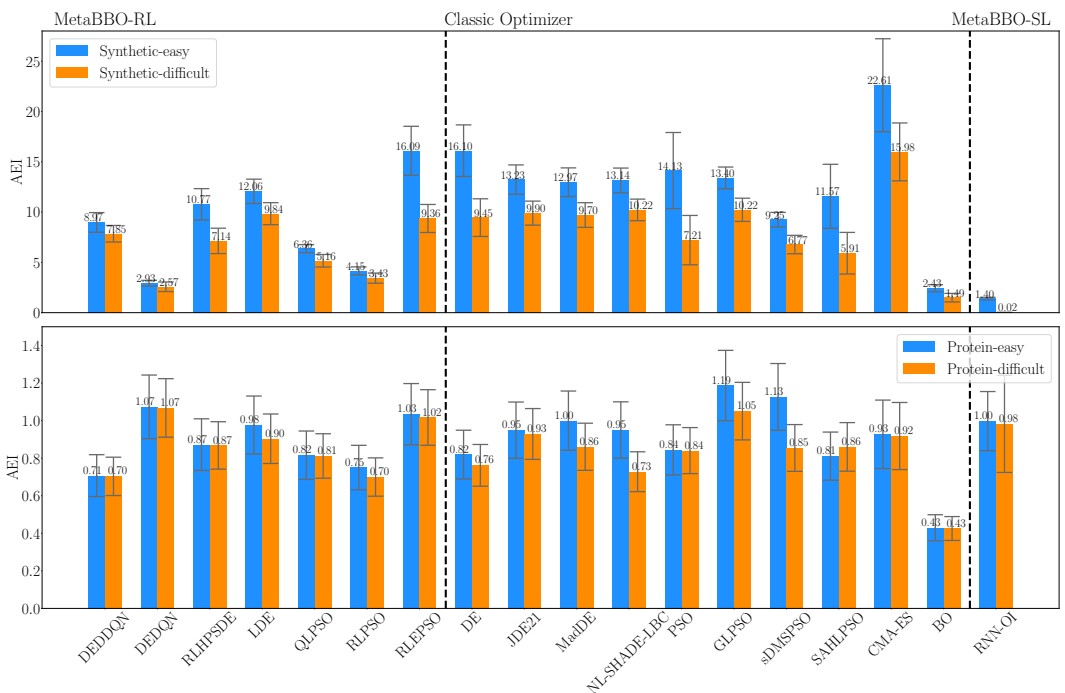

Figure 4: AEI scores of baselines, with error bars denoting the robustness across different tasks. **Top:** Results on Synthetic testsuites with *easy* or *difficult* difficulty. **Bottom:** Results on Protein-Docking testsuites with *easy* or *difficult* difficulty.

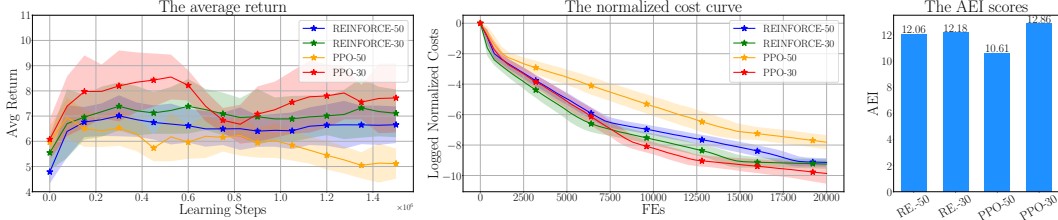

Figure 5: Hyper-tuning for the MetaBBO-RL approach LDE [21] using a $2 \times 2$ grid search. **Left:** The average return during the training over 10 trials. **Middle:** The normalized cost over 51 independent runs during the test. **Right:** The corresponding AEI scores during the test.

optimizers when tested on different problem sets. For example, DEDQN generally performs the best for the Protein-Docking problems, ranking first and third in *difficult* and *easy* modes, respectively. This suggests that the classic optimizers might be only tailored to a specific family of problems, while MetaBBO-RL learns to enhance the low-level optimizer through meta-level learning, thereby exhibiting stronger robustness. Due to the space limitation, we leave additional comparison results, including those on other testsuites, as well as the detailed sub-metrics such as running time, algorithm complexity, and per-instance optimization results in Appendix D.

### 4.3 Hyper-tuning a MetaBBO-RL approach

Tuning a MetaBBO-RL requires efforts in configuring both the meta-level RL *Agent* and the low-level black-box *Optimizer*. Fortunately, our MetaBox provides *Template coding* with convenient interfaces to assist users in accomplishing this task. To illustrate this, we conduct a $2 \times 2$ grid search to hyper-tune the MetaBBO-RL approach LDE [21]. We noticed that LDE achieved lower AEI scores on Synthetic-easy testsuites than some classic optimizers such as JDE21 and MadDE (see Figure 4). In our study, we investigate both REINFORCE [51] and PPO [52] RL agents for LDE, setting the population size (a hyper-parameter of DE) to either 30 or 50 (where the original version of LDE uses

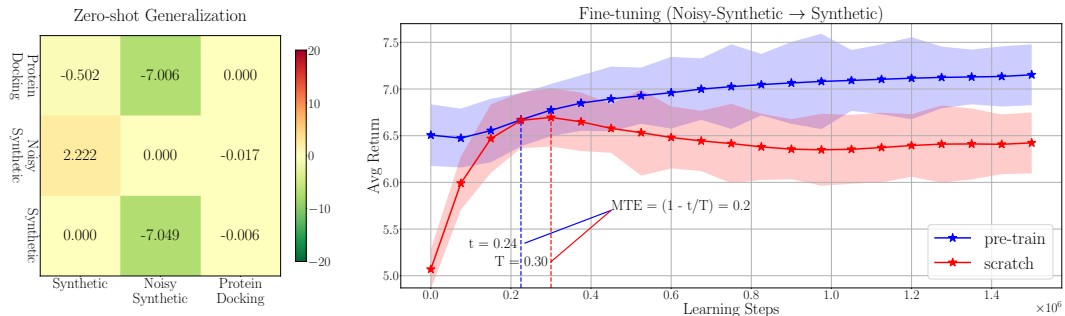

Figure 6: Meta performance (MGD and MTE) of LDE [21] across different tasks, tested under the *easy* mode. **Left:** Logits on $i$-th row and $j$-th column is the $MGD(i, j)$, the smaller the better. **Right:** The average return over 10 trials is compared between the LDE models pre-trained and trained from scratch. The annotated MTE score shows that it saves 0.2x learning steps through fine-tuning.

REINFORCE and a population size of 50). Figure 5 depicts the average return along the training, the optimization curve during testing, and the AEI scores of these four LDE versions. It turns out that LDE with the PPO agent and a population size of 30 achieves higher scores and surpasses JDE21 and MadDE. This shows that, by leveraging the capabilities of MetaBox and conducting hyper-tuning studies, researchers are able to pursue the best configurations for their MetaBBO-RL methods.

### 4.4 Investigating generalization and transfer learning performance

Existing MetaBBO-RL methods [19–25] rarely assess their generalization and transfer ability, although they are two crucial aspects for evaluating the effectiveness of a learning-based method. To bridge this gap, we have introduced two explicit indicators (namely the MGD and MTE) in MetaBox to measure the level of learning achieved by a MetaBBO-RL algorithm. In Figure 6, we display the values of MGD and MTE for the MetaBBO-RL approach LDE.

**Generalization performance.** Results in the MGD plot (the left part of Figure 6) show that the pre-trained LDE model possesses a strong generalization ability. When the model is pre-trained on the Noisy-Synthetic testsuites, it exhibits a positive MGD on the Synthetic testsuites, with only a 2.222% drop in AEI, which is considered acceptable. Additionally, the LDE model trained on the Protein-Docking-easy or Synthetic-easy testsuites both exhibits exceptional generalization to the Noisy-Synthetic-easy testsuites, as evidenced by an MGD of $-7.006\%$ and $-7.049\%$. Such robust generalization abilities are highly desirable in practical applications.

**Transfer performance.** We fine-tune the LDE pre-trained on the Noisy-Synthetic testsuites for the Synthetic testsuites. The progress of this fine-tuning process is depicted in the MTE plot (the right part of Figure 6), showing the average return over 10 trials. The results reveal that the pre-trained LDE requires fewer learning steps to arrive at the peak return level than the LDE trained from scratch, with the MTE value of 20%. Compared with the MGD value of 2.222% when zero-shotting the pre-trained LDE for the Synthetic testsuites, it can be noticed that the MGD and MTE capture two different aspects of the learning effectiveness. By considering both measures, researchers can gain deeper insights into the learning behavior and effectiveness of their MetaBBO-RL approaches.

## 5 Discussion and Future Work

We propose MetaBox, a benchmark platform for Meta-Black-Box Optimization with Reinforcement Learning (MetaBBO-RL). It intends to, 1) provide the first unified benchmark platform, 2) simplify coding towards efficient researching, 3) provide broad testsuites and baselines for comprehensive comparison, and 4) provide novel evaluation metrics for in-depth analysis.

Our preliminary experimental investigations lead to the following key observations. First, although hand-crafted optimizers currently outperform learning-based optimizers (e.g., MetaBBO-RL), they are mostly designed to specific problem sets and become less effective when applied to other problem domains. In contrast, MetaBBO-RL exhibits adaptability across various optimization scenarios,

showing stronger robustness across different problem domains. This is a significant advantage, as it allows MetaBBO-RL to perform well on diverse problem sets without the need for much manual customization. Second, there is still room for further improvement in MetaBBO-RL, by discovering more effective designs in both meta-level agents and low-level optimizers. We find that even basic modifications, such as incorporating a PPO agent and adjusting the population size, can bring a significant performance boost. Third, interpreting the generalization and transfer effects in MetaBBO-RL can be challenging. While we were able to observe positive and negative effects in terms of MGD and MTE, rationally interpreting and understanding them is still difficult. Hence, additional studies are needed to interpret the learning effects of MetaBBO-RL in a more comprehensible manner.

While our MetaBox presents a significant advancement, we also acknowledge its potential limitations. It is important to note that the evaluation of BBO performance is not a one-size-fits-all endeavour. Different practical applications may have varying preferences and additional concerns, such as solution robustness, solution diversity, parallelization and scalability. These nuances extend beyond the scope of a single evaluation metric. In conjunction with the proposed AEI, MetaBox is committed to exploring more pragmatic and advanced evaluation criteria. Moreover, the problem sets in MetaBox will be expanded to include more diverse and eye-catching BBO tasks. Additionally, proactive maintenance and updates to MetaBox, including the extension of the baseline library, will also be pursued to ensure it is up-to-date with the latest methodologies in the field.

## Acknowledgments and Disclosure of Funding

This work was supported in part by the National Natural Science Foundation of China under Grant 62276100, in part by the Guangdong Natural Science Funds for Distinguished Young Scholars under Grant 2022B1515020049, in part by the Guangdong Regional Joint Funds for Basic and Applied Research under Grant 2021B1515120078, and in part by the TCL Young Scholars Program.

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

# MetaBox: A Benchmark Platform for Meta-Black-Box Optimization with Reinforcement Learning (Appendix)

## A Details of Testsuites

### A.1 Synthetic

There are 24 functions in the Synthetic testsuites. These noiseless synthetic functions, established by Hansen et al. [47], are taken from COCO [28]. The mathematical definitions and formulations of these functions can be found in Hansen et al. [47]. We have listed these functions in Table 2. To provide different levels of difficulty, we have divided the 24 functions into two parts: 25% and 75%. The 25% portion includes functions No.1, 5, 6, 10, 15, and 20, while the remaining 18 functions make up the 75% portion. In the *easy* mode, the 75% portion is used for training, and the 25% portion is used for testing. On the other hand, in the *difficult* mode, the 25% portion is used for training, and the 75% portion is used for testing. The maximum number of function evaluations (*maxFEs*) for problems in these test suites is set to $2 \times 10^4$.

Table 2: Summary of the Synthetic test functions.

| | No. | Functions | $f_i^*$ |
|---|---|---|---|
| Separable functions | 1 | Sphere Function | 700 |
| | 2 | Ellipsoidal Function | 1900 |
| | 3 | Rastrigin Function | 100 |
| | 4 | Buche-Rastrigin Function | 1000 |
| | 5 | Linear Slope | 2000 |
| Functions with low or moderate conditioning | 6 | Attractive Sector Function | 400 |
| | 7 | Step Ellipsoidal Function | 300 |
| | 8 | Rosenbrock Function, original | 1300 |
| | 9 | Rosenbrock Function, rotated | 1300 |
| Functions with high conditioning and unimodal | 10 | Ellipsoidal Function | 600 |
| | 11 | Discus Function | 900 |
| | 12 | Bent Cigar Function | 2000 |
| | 13 | Sharp Ridge Function | 2400 |
| | 14 | Different Powers Function | 1500 |
| Multi-modal functions with adequate global structure | 15 | Rastrigin Function (non-separable counterpart of F3) | 1700 |
| | 16 | Weierstrass Function | 1400 |
| | 17 | Schaffers F7 Function | 200 |
| | 18 | Schaffers F7 Function, moderately ill-conditioned | 700 |
| | 19 | Composite Griewank-Rosenbrock Function F8F2 | 1700 |
| Multi-modal functions with weak global structure | 20 | Schwefel Function | 2100 |
| | 21 | Gallagher's Gaussian 101-me Peaks Function | 700 |
| | 22 | Gallagher's Gaussian 21-hi Peaks Function | 2400 |
| | 23 | Katsuura Function | 600 |
| | 24 | Lunacek bi-Rastrigin Function | 1200 |
| Default search range: $[-5, 5]^D$ | | | |

### A.2 Noisy-Synthetic

There are 30 functions in the Noisy-Synthetic testsuites. These noiseless synthetic functions, established by Hansen et al. [47], are taken from COCO [28]. The mathematical definitions and formulations of these functions can be found in Hansen et al. [47]. We have listed these functions in

Table 2. To provide different levels of difficulty, we have divided the 24 functions into two parts: 25% and 75%. The 25% portion includes functions No.1, 5, 15, 16, 17, 19, 20, 25, while the remaining 22 functions make up the 75% portion. The maximum number of function evaluations (*maxFEs*) for problems in these test suites is set to $2 \times 10^4$.

Table 3: Summary of the Noisy-Synthetic test functions.

| | No. | Functions | $f_i^*$ |
|---|---|---|---|
| Functions with moderate noise | 1 | Sphere with moderate gaussian noise | 700 |
| | 2 | Sphere with moderate uniform noise | 1900 |
| | 3 | Sphere with moderate seldom cauchy noise | 100 |
| | 4 | Rosenbrock with moderate gaussian noise | 1000 |
| | 5 | Rosenbrock with moderate uniform noise | 2000 |
| | 6 | Rosenbrock with moderate seldom cauchy noise | 400 |
| Functions with severe noise | 7 | Sphere with gaussian noise | 100 |
| | 8 | Sphere with uniform noise | 2000 |
| | 9 | Sphere with seldom cauchy noise | 2000 |
| | 10 | Rosenbrock with gaussian noise | 500 |
| | 11 | Rosenbrock with uniform noise | 400 |
| | 12 | Rosenbrock with seldom cauchy noise | 700 |
| | 13 | Step ellipsoid with gaussian noise | 1900 |
| | 14 | Step ellipsoid with uniform noise | 800 |
| | 15 | Step ellipsoid with seldom cauchy noise | 2300 |
| | 16 | Ellipsoid with gaussian noise | 200 |
| | 17 | Ellipsoid with uniform noise | 100 |
| | 18 | Ellipsoid with seldom cauchy noise | 200 |
| | 19 | Different Powers with gaussian noise | 500 |
| | 20 | Different Powers with uniform noise | 1600 |
| | 21 | Different Powers with seldom cauchy noise | 1000 |
| Highly multi-modal functions with severe noise | 22 | Schaffer's F7 with gaussian noise | 2400 |
| | 23 | Schaffer's F7 with uniform noise | 1400 |
| | 24 | Schaffer's F7 with seldom cauchy noise | 2400 |
| | 25 | Composite Griewank-Rosenbrock with gaussian noise | 2200 |
| | 26 | Composite Griewank-Rosenbrock with uniform noise | 400 |
| | 27 | Composite Griewank-Rosenbrock with seldom cauchy noise | 2500 |
| | 28 | Gallagher's Gaussian Peaks 101-me with gaussian noise | 1000 |
| | 29 | Gallagher's Gaussian Peaks 101-me with uniform noise | 600 |
| | 30 | Gallagher's Gaussian Peaks 101-me with seldom cauchy noise | 2100 |
| Default search range: $[-5, 5]^D$ | | | |

### A.3 Protein-Docking

Ab initio protein docking poses a significant challenge in optimizing a noisy and costly *black-box* function [56]. The objective of ab initio protein docking, for a fixed basis protein conformation $x_0$, is to minimize the change in Gibbs free energy resulting from protein interaction between $x_0$ and any other conformation $x$, denoted as $E_{int}(x, x_0)$. Following the approach of Moal and Bates [57], we formulate the objective as follows:

$$\min_x E_{int}(x, x_0) = \sum_i^{atoms} \sum_j^{atoms} E(x^i, x_0^j), \tag{6}$$

where $E(x^i, x_0^j)$ is the energy between any pair atoms of $x$ and $x_0$, and is defined as :

$$E_{i,j} = \begin{cases} \frac{q_i q_j}{\epsilon r_{i,j}} + \sqrt{\epsilon_i \epsilon_j} \left[ (\frac{R_{i,j}}{r_{i,j}})^{12} - (\frac{R_{i,j}}{r_{i,j}})^6 \right], & \text{if } r_{i,j} < 7 \\ \left[ \frac{(r_{off} - f_{i,j})^2 (r_{off} + 2r_{i,j} - 3r_{on})}{(r_{off} - r_{on})^3} \right] \left\{ \frac{q_i q_j}{\epsilon r_{i,j}} + \sqrt{\epsilon_i \epsilon_j} \left[ (\frac{R_{i,j}}{r_{i,j}})^{12} - (\frac{R_{i,j}}{r_{i,j}})^6 \right] \right\}, & \text{if } 7 \leq r_{i,j} \leq 9 \\ 0 & \text{if } r_{i,j} > 9 \end{cases} \tag{7}$$

All parameters and calculations are taken from the Charmm19 force field [58]. A switching function is employed, between 7 and 9, to disregard long-distance interaction energy. To construct the task

instance $x_0$ for protein docking, we select 28 protein-protein complexes from the protein docking benchmark set 4.0 [29]. Each complex is associated with 10 different starting points, chosen from the top-10 start points identified by ZDOCK [59]. Consequently, the Protein-Docking testsuites comprise a total of 280 docking task instances. We split these instances into two parts, with approximately 75% and 25% of the instances serving as the *difficulty* levels within the testsuites. It is important to note that we parameterize the search space as $\mathbb{R}^{12}$, which is a reduced dimensionality compared to the original protein complex [56]. This reduction in dimensionality enables computational time savings while retaining the optimization nature of the problem. Due to the computationally expensive evaluations, the maximum number of function evaluations (*maxFEs*) is set to $10^3$. The default search range for the optimization is $[-\infty, \infty]$.

# B    Details of AEI Calculation

We describe the calculations of AEI as follows. Suppose we want to test a (Meta)BBO approach $\Lambda$ to obtain its AEI on Testsuites $A$, i.e., AEI($\Lambda$,$A$). Suppose $A$ has $K$ problem instances, and we run $\Lambda$ to optimize each problem in $A$ for $N$ times. In this testing process, we record 3 kinds of values:

**Best objective value.** The best objective value $v_{obj}^{k,n}$ denotes the best objective found by $\Lambda$ on the $k$-th problem during the $n$-th test run. To convert its monotonicity, we conduct an inverse transformation, i.e., $v_{obj}^{k,n} = \frac{1}{v_{obj}^{k,n}}$.

**Consumed function evaluations.** We let $v_{fes}^{k,n}$ denote the number of consumed function evaluations when the approach $\Lambda$ terminates the optimization process for the $k$-th problem during the $n$-th test run, where $\Lambda$ would terminate the optimization either upon locating a high-quality solution that fulfils a predetermined accuracy or when it reaches the maximum allowed function evaluations, i.e., *maxFEs*. We then pre-process this value by a min-max conversion for the consistency towards AEI maximization, i.e., $v_{fes}^{k,n} = \frac{maxFEs}{v_{fes}^{k,n}}$.

**Runtime complexity.** The calculation includes three steps following the conventions of IEEE CEC Competitions [34, 55]. We first pre-calculate a $T_0$ which is a basic normalization term, calculating the computation time of some basic *numpy* operations (e.g., add, division, log, exp) as references. We record the time $T_1^{k,n}$ for $\Lambda$ to evaluate the $k$-th problem's objective during $n$-th run, and $T_2^{k,n}$ as total running time for $\Lambda$ to solve the $k$-th problem's objective during $n$-th run. The runtime complexity $v_{com}^{k,n}$ is then calculated by $\frac{T_2^{k,n} - T_1^{k,n}}{T_0}$. Then we pre-process this value by a min-max conversion for consistency towards AEI maximization, i.e., $v_{obj}^{k,n} = \frac{1}{v_{obj}^{k,n}}$.

# C    Baseline Setup

**Classic optimizer.**  Random Search does not necessitate any control parameters. We establish the control parameters for CMA-ES, DE, and PSO in alignment with DEAP [53], while the control parameters for BO are set in accordance with Scikit-Optimizer [54]. For the classic optimizers implemented by us, the control parameters are delineated and can be accessed at https://github.com/GMC-DRL/MetaBox/blob/main/src/control_parameters_classic.md.

**MetaBBO-SL.** We briefly introduce the control parameters of *RNN-OI* here. It adopts an LSTM network to generate the next solution position on the observation of some present optimization status. This LSTM has *input size* as $problem\_dim + 2$, *hidden size* as 32, and *projection size* as $problem\_dim$. The *Adam* optimizer is used to train the network with a fixed learning rate $10^{-5}$. However, it can only optimize a problem for around $maxFEs = 100$ due to the inefficiency arising from utilizing RNNs to model an extended optimization horizon.

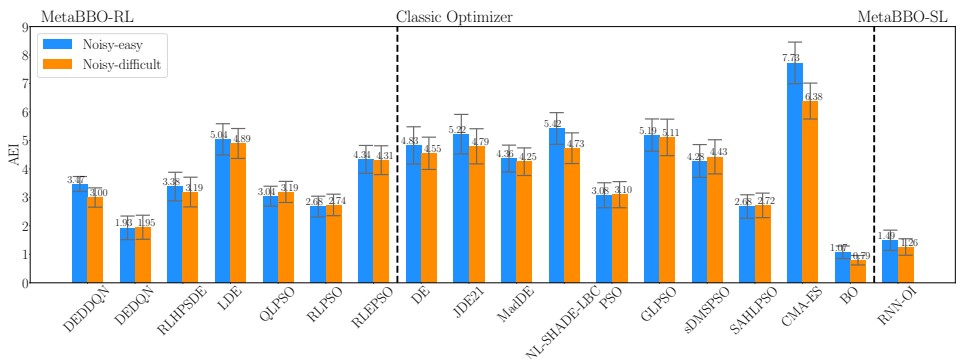

Figure 7: AEI scores with error bars of baselines on Noisy-Synthetic testsuites.

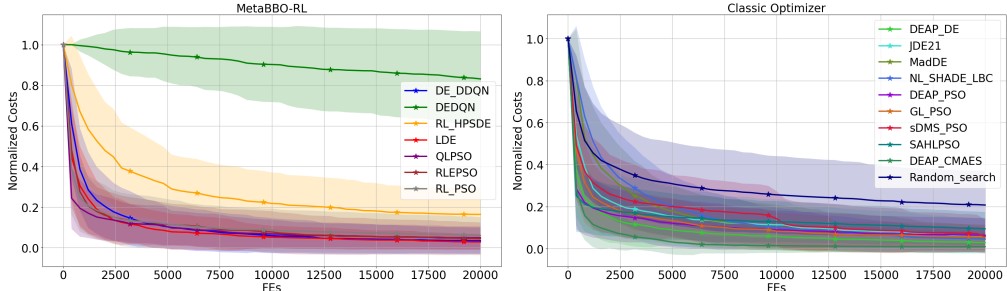

Figure 8: Average cost curve of baselines over Noisy-Synthetic-easy testsuites.

# D Additional Results and Comparisons

## D.1 Additional results

In addition to the AEI scores, MetaBox provides additional experimental results that complement the comprehensive analysis of MetaBBO-RL approaches. These results, along with AEI, are automatically generated and presented after the testing process. The currently available results for all baselines in *Baseline Library* can be accessed at https://github.com/GMC-DRL/MetaBox/blob/main/post_processed_data/content.md. We briefly preview these additional results here:

**Overall performance table.** For each baseline in the *Baseline Library*, it presents the average final objective value (*Obj*), the average *Gap* with respect to the performance of CMA-ES, and the average consumed function evaluation times (*FEs*). These values are obtained based on 51 independent runs.

**Wall-time table.** For each baseline in the *Baseline Library*, it presents the average running time $\frac{1}{K \times N} \sum_{k=1}^{K} \sum_{n=1}^{N} T_2^k$ and the average algorithm complexity Z-score $\frac{1}{K} \sum_{k=1}^{K} Z_{com}^k$. These values are averaged over all the problems in the testsuites and 51 independent runs.

**Cost curve.** For each baseline in the *Baseline Library*, the figure draws the averaged optimization curves (cost curves) over all the problems in the testsuites and 51 independent runs.

In this appendix, as preview examples, we present the experimental results of baselines on the Noisy-Synthetic testsuites, which includes AEI scores with error bars on both the Noisy-Synthetic-easy and the Noisy-Synthetic-difficult testsuites in Figure 7, the overall performance table on Noisy-Synthetic-easy testsuites in Table 4, the wall-time table on Noisy-Synthetic-easy testsuites in Table 5, and the cost curves of MetaBBO-RL baselines on Noisy-Synthetic-easy testsuites in Figure 8. Note that they are partially displayed due to space limitations, without loss of generality.

## D.2 Investigation on RLEPSO

**Hyper-tuning.** Since RLEPSO [25] also shows competitive performance in all of the three testsuites, we execute the same $2 \times 2$ grid-search experiment for it and measure its MGD and MTE. Similarly, we

Table 4: Per-instance optimization results of baselines on Noisy-Synthetic-easy testsuites, showing the first 3 problems in the test set for illustration purpose.

| | | Sphere-moderate-gauss | | | Rosenbrock-moderate-uniform | | | Step-Ellipsoidal-cauchy | | |
|---|---|---|---|---|---|---|---|---|---|---|
| | | Obj | Gap | FEs | Obj | Gap | FEs | Obj | Gap | FEs |
| | Random | 1.101E+1 (2.739E+0) | 1.000 | 2.000E+4 (0.000E+0) | 2.267E+3 (1.023E+3) | 1.000 | 2.000E+4 (0.000E+0) | 6.363E+2 (2.709E+2) | 1.000 | 2.000E+4 (0.000E+0) |
| **Classic** | CMA-ES | 7.689E-9 (1.844E-9) | 0.000 | 4.958E+3 (1.931E+2) | 1.147E+0 (2.696E+0) | 0.000 | 1.624E+4 (2.174E+3) | 5.743E+0 (4.048E+1) | 0.000 | 6.119E+3 (3.629E+3) |
| | PSO | 1.906E+0 (8.624E-1) | 0.173 | 2.000E+4 (0.000E+0) | 2.679E+2 (5.578E+2) | 0.118 | 2.000E+4 (0.000E+0) | 2.950E+2 (2.453E+2) | 0.459 | 2.000E+4 (0.000E+0) |
| | SAHLPSO (-g) | 4.477E+0 (2.875E+0) | 0.407 | 2.000E+4 (0.000E+0) | 9.651E+2 (9.798E+2) | 0.426 | 2.000E+4 (0.000E+0) | 4.189E+2 (2.584E+2) | 0.655 | 2.000E+4 (0.000E+0) |
| | sDMSPSO (v1) | 1.661E+0 (7.380E-1) | 0.151 | 2.010E+4 (0.000E+0) | 1.080E+2 (4.728E+1) | 0.047 | 2.010E+4 (0.000E+0) | 1.395E+2 (1.922E+2) | 0.212 | 2.010E+4 (0.000E+0) |
| | GLPSO | 1.292E-6 (7.497E-7) | 0.000 | 2.000E+4 (0.000E+0) | 6.867E+0 (8.194E-1) | 0.003 | 2.000E+4 (0.000E+0) | 3.322E+2 (3.195E+2) | 0.518 | 2.000E+4 (0.000E+0) |
| | DE (rand/1) | 8.010E-9 (1.636E-9) | 0.000 | 4.601E+3 (1.794E+2) | 5.719E+0 (1.373E+0) | 0.002 | 2.000E+4 (0.000E+0) | 1.373E+2 (2.041E+2) | 0.209 | 2.000E+4 (0.000E+0) |
| | JDE21 | 5.302E-9 (2.477E-9) | -0.000 | 6.283E+3 (1.562E+3) | 6.212E+0 (5.923E+0) | 0.002 | 2.001E+4 (0.000E+0) | 2.227E+2 (2.440E+2) | 0.344 | 2.001E+4 (0.000E+0) |
| | NL-SHADE-LBC | 7.826E-9 (1.494E-9) | 0.000 | 1.444E+4 (1.161E+2) | 4.541E+0 (8.124E-1) | 0.001 | 2.000E+4 (0.000E+0) | 1.849E+2 (2.153E+2) | 0.284 | 2.000E+4 (0.000E+0) |
| | MadDE | 7.988E-9 (1.602E-9) | 0.000 | 1.945E+4 (1.438E+2) | 5.361E+0 (1.083E+0) | 0.002 | 2.000E+4 (0.000E+0) | 1.613E+2 (2.382E+2) | 0.247 | 2.000E+4 (0.000E+0) |
| | Bayesian (BO) | 9.144E-2 (1.197E-1) | 0.008 | 1.000E+2 (0.000E+0) | 4.506E+3 (1.303E+4) | 1.988 | 1.000E+2 (0.000E+0) | 1.024E+3 (2.678E+1) | 1.615 | 1.000E+2 (0.000E+0) |
| **Learnable** | QLPSO | 2.568E+0 (2.915E+0) | 0.233 | 2.000E+4 (0.000E+0) | 1.467E+2 (2.788E+2) | 0.064 | 2.000E+4 (0.000E+0) | 1.411E+2 (2.387E+2) | 0.215 | 2.000E+4 (0.000E+0) |
| | RLPSO | 2.419E+0 (1.277E+0) | 0.220 | 2.000E+4 (0.000E+0) | 1.607E+2 (1.164E+2) | 0.070 | 2.000E+4 (0.000E+0) | 1.440E+2 (1.843E+2) | 0.219 | 2.000E+4 (0.000E+0) |
| | RLEPSO | 7.521E-6 (1.163E-5) | 0.000 | 2.000E+4 (1.707E+0) | 7.244E+0 (2.131E+0) | 0.003 | 2.000E+4 (1.473E+0) | 1.580E+2 (1.824E+2) | 0.241 | 2.000E+4 (1.695E+0) |
| | DEDQN | 3.369E+1 (9.025E+0) | 3.060 | 2.000E+4 (0.000E+0) | 2.152E+4 (9.820E+3) | 9.498 | 2.000E+4 (0.000E+0) | 1.097E+3 (4.189E+1) | 1.730 | 2.000E+4 (0.000E+0) |
| | DEDDQN | 7.907E-9 (1.523E-9) | 0.000 | 1.692E+4 (5.373E+2) | 2.913E+0 (2.859E+0) | 0.001 | 1.977E+4 (7.700E+2) | 1.022E+2 (1.726E+2) | 0.153 | 2.000E+4 (0.000E+0) |
| | LDE | 8.127E-9 (1.628E-9) | 0.000 | 9.685E+3 (3.920E+2) | 2.681E+0 (2.639E+0) | 0.001 | 2.000E+4 (0.000E+0) | 1.292E+2 (2.007E+2) | 0.196 | 2.000E+4 (0.000E+0) |
| | RLHPSDE | 1.118E-1 (6.738E-2) | 0.010 | 2.025E+4 (0.000E+0) | 4.248E+1 (2.126E+1) | 0.018 | 2.025E+4 (0.000E+0) | 6.269E+2 (3.087E+2) | 0.985 | 2.025E+4 (0.000E+0) |
| | RNN-OI | 6.940E+1 (6.824E-1) | 6.304 | 1.000E+2 (0.000E+0) | 3.177E+4 (6.347E+2) | 14.021 | 1.000E+2 (0.000E+0) | 1.099E+3 (1.198E+1) | 1.734 | 1.000E+2 (0.000E+0) |

Table 5: Running time and algorithm complexity of baselines on Noisy-Synthetic-easy testsuits.

| Baselines | CMA-ES | PSO | SAHLPSO | sDMSPSO | GLPSO | DE |
|---|---|---|---|---|---|---|
| **T2 (ms)** | 3.337E+02 | 2.021E+03 | 4.506E+03 | 6.484E+01 | 5.508E+01 | 8.838E+02 |
| **Z-score** | 9.540E-01 | 8.093E-01 | 7.254E-01 | 1.047E+00 | 1.066E+00 | 8.420E-01 |
| **Baselines** | **JDE21** | **NL-SHADE-LBC** | **MadDE** | **QLPSO** | **RLPSO** | **RLEPSO** |
| **T2 (ms)** | 1.639E+02 | 2.036E+02 | 4.015E+02 | 4.442E+03 | 1.337E+04 | 3.540E+02 |
| **Z-score** | 9.588E-01 | 9.370E-01 | 8.827E-01 | 7.265E-01 | 6.530E-01 | 8.918E-01 |
| **Baselines** | **DEDQN** | **DEDDQN** | **LDE** | **RLHPSDE** | **RNN-OI** | **BO** |
| **T2 (ms)** | 2.309E+02 | 2.787E+04 | 3.303E+02 | 5.867E+02 | 4.846E+03 | 1.407E+07 |
| **Z-score** | 9.262E-01 | 6.101E-01 | 8.972E-01 | 8.517E-01 | 7.216E-01 | 3.542E-01 |

investigate the combination of different RL methods (REINFORCE or PPO) and different population sizes (100 or 50). The original RLEPSO is implemented in PPO with 100 population size. The grid search results are shown in Figure 9. The RLEPSO with the PPO agent and a population size of 50 has the highest average return during the training, however, this configuration shows the worst optimization results (considering the normalized cost) during the test. On the contrary, RLEPSO with the REINFORCE agent and a population size of 50 has the lowest average return during the training and the lowest normalized cost during the test. This may indicate that the design of the RLEPSO reward function may not be appropriate. Meanwhile, we note that for RLEPSO, changing its RL agent from PPO to REINFORCE may slightly improve the overall performance.

**Generalization performance.** We also present the MGD of RLEPSO on the testsuites with an *easy* difficulty level, displayed on the left side of Figure 10. It appears that RLEPSO exhibits poor generalization ability. For instance, when the RLEPSO model is pre-trained on the Protein-Docking-easy testsuites, there is a significant 40.312% drop in the AEI score. These results indicate that while RLEPSO achieves competitive AEI performance, its generalization ability is much worse compared

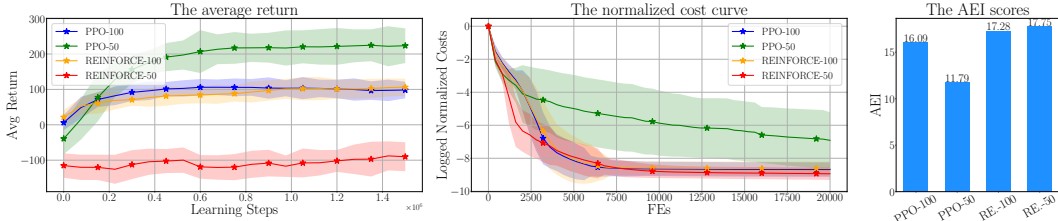

Figure 9: Hyper-tuning for the MetaBBO-RL approach RLEPSO [25] using a $2 \times 2$ grid search. **Left:** The average return during the training over 10 trials. **Middle:** The normalized cost over 51 independent runs during the test. **Right:** The corresponding AEI scores during the test.

to LDE. The subpar MGD performance of RLEPSO may be attributed to its non-generalizable state design, which directly utilizes the consumed FEs as the state representation.

**Transfer performance.** We performed fine-tuning on the RLEPSO model that was pre-trained on the Noisy-Synthetic-easy testsuites for solving the Synthetic-easy testsuites. The progress of the average return over 10 trials is shown on the right side of Figure 10. However, the pre-trained RLEPSO exhibits negative transferability, with an MTE value of 0. This transfer failure suggests that some of the current MetaBBO-RL methods face overfitting issues, which in the case of RLEPSO may be attributed to its oversimplified state representation or other design elements that lack generalizability.

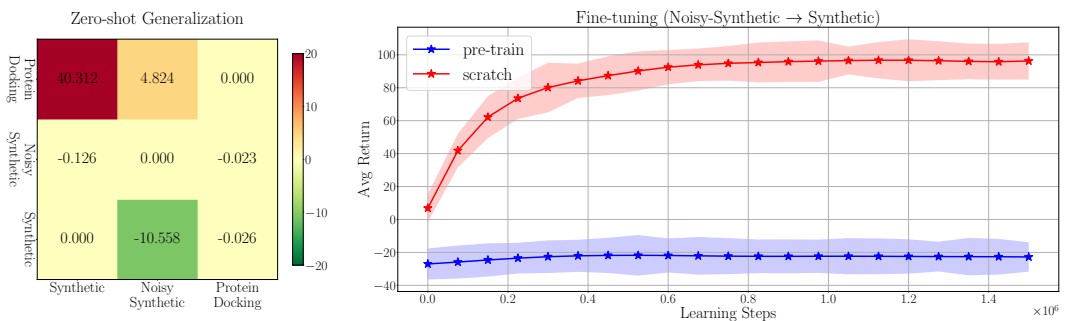

Figure 10: Meta performance (MGD and MTE) of RLEPSO [25] across different tasks, tested under the *easy* mode. **Left:** Logits on $i$-th row and $j$-th column is the $MGD(i, j)$, the smaller the better. **Right:** the average return over 10 trials is compared between the RLEPSO models pre-trained and trained from scratch, with a $MTE = 0$. This is a transfer failure.

# E Used Assets

MetaBox is an open-sourced tool, and it can be accessed at: https://github.com/GMC-DRL/MetaBox. It is currently licensed under the BSD 3-Clause License. In Table 6, we provide a list of the resources or assets utilized in MetaBox, along with their respective licenses. It is important to note that we adhere strictly to these licenses during the development of MetaBox.

Table 6: Used assets and their licenses

| Type | Asset | Codebase | License |
|------|-------|----------|---------|
| Baseline | CMA-ES [5] | DEAP [53] | LGPL-3.0 License |
| | DE [10] | DEAP [53] | LGPL-3.0 License |
| | PSO [8] | DEAP [53] | LGPL-3.0 License |
| | BO [14] | Scikit-Optimizer [54] | BSD-3-Clause License |
| Testsuites | Synthetic | COCO [28] | BSD 3-clause License |
| | Noisy-Synthetic | COCO [28] | BSD 3-clause License |
| | Protein-Docking | Protein-Docking 4.0 [29] | - |

---
**Algorithm 1:** MetaBoxTrainer
---
**Input:** User-designed learnable agent $A$, User-designed optimizer $O$, User-specified problem set $P_{\text{train}}$ with configuration $C_{\text{train}}$

**Output:** Trained Agent $A$, training records

Initialize problem set $P_{\text{train}}$ with configuration $C_{\text{train}}$ as the training dataset;

**while** *max learning steps Not reached* **do**
    **for** *each problem instance ins $\in P_{train}$* **do**
        Construct optimization environment $Env = \text{Env\_Construction}(O, ins)$;
        $A.\text{train\_episode}(Env)$;
        Record training data;
        Plot training figures;
    **end**
**end**

Summarize and visualize the training records in $Logger$ and return the trained agent;

---

---
**Algorithm 2:** TrainingEpisode
---
**Input:** User-designed learnable agent $A$, Constructed environment $Env$

**Output:** Training records

$state = Env.\text{reset}()$;

**while** *termination condition Not achieved* **do**
    $action = A.\text{get\_action}(state)$;
    $next\_state, reward, info = Env.\text{step}(action)$;
    Store transition $<state, action, reward, next\_state>$;
    Update agent $A$;
    Record training data and plot figures;
    $state = next\_state$;
**end**

Summarize training records and return;

---

## F Pseudo-code of MetaBox

The pseudo-code of Trainer is shown in Algorithm 1. Given the user-designed learnable agent $A$ and optimizer $O$, our MetaBox framework firstly initializes the problem instance set (e.g., determines problem types, indexes file directories and loads data to construct problem instances according to the user-specified configuration $C_{\text{train}}$). Then MetaBox iteratively performs training on each training instance until the max learning step is reached. For each instance, a Gym-style environment $Env$ is constructed to merge $O$ and the problem instance together, so as to provide a unified interface. Agent $A$ then calls the train_episode() interface for interacting with $Env$ and performing the actual training. All the generated logs during training are managed by the $Logger$ as depicted in Figure 1.

We now turn to the details of train_episode(). Given the variance among numerous learning approaches, such as RL and SL, the internal workflow of train_episode() necessitates implementation aligned with specific designs. Within MetaBox, we offer a range of examples through the implementation of various baselines, serving as guides for users who wish to develop their own interfaces. Meanwhile, we note that our baseline library also covers the implementations of prevalent RL algorithms like PPO and REINFORCE. In Algorithm 2, we showcase a straightforward example of the workflow for implementing RL training algorithms within train_episode(). Starting from the $Env$ initialization, in each step, the agent $A$ provides $Env$ the action according to the state, receives the next state, reward and other information, and uses them to update the policy. Within the env.step() interface, the action is translated into configurations which are then applied to the optimizer $O$. The optimizer then performs update rules to derive and evaluate new solutions. Following this, rewards and subsequent states are computed, with certain logging information being concurrently summarized.

As for the Tester shown in Algorithm 3, it first initializes the test problem set which is used to evaluate each algorithm in an algorithm set (including several baseline agents for comparison and the user's trained agent). For learning-based algorithms, respective agents and optimizers are then

**Algorithm 3:** MetaBoxTester

---

**Input:** User-specified algorithm set $B$ including baselines and user's trained agent,
       User-specified problem set $P_{\text{test}}$ with configuration $C_{\text{test}}$
**Output:** Testing results
Initialize problem set $P_{\text{test}}$ with configuration $C_{\text{test}}$ as the test set;
**for** *each algorithm* $alg \in B$ **do**
    **for** *each problem instance* $ins \in P_{test}$ **do**
        **if** $alg$ *is a learning-based method* **then**
            | Initialize $alg.agent$ and $alg.optimizer$;
        **end**
        **else**
            | Initialize $alg.optimizer$ as an optimizer;
        **end**
        Construct optimization environment $Env = $ Env_Construction$(alg.optimizer, ins)$;
        $alg$.rollout_episode$(Env)$;
        Record testing data;
        Plot testing figures;
    **end**
**end**
Summarize testing results and call $Logger$ for standardized metrics and visualization;

---

initialized, which may involve loading network models and initializing parameters. Conversely, classic optimizers are simply initialized as the optimizers. The rollout_episode() interface mirrors the train_episode() but removes the policy update procedures. After the testing, the recorded testing logs will be organized and presented within the $Logger$.

