# OpenReview forum: "MetaBox: A Benchmark Platform for Meta-Black-Box Optimization with Reinforcement Learning"
_NeurIPS.cc/2023/Track/Datasets_and_Benchmarks — NeurIPS 2023 Datasets and Benchmarks Oral_

### Official Review · Reviewer_t7Zi · 2023-07-16

**Rating:** 7
**Confidence:** 3
**Correctness:** Yes
**Clarity:** Yes

**Strengths:**

1. The benchmark is comprehensive, with numerous instances and baselines for comparison.
2. The documentation (README) provides sufficient details for users to use the package.
3. The benchmark provides a standardized way to evaluate the methods in this domain.

**Additional Feedback:**

The benchmark is well-developed. The task is similar to hyperparameter search and BBO. Since there are already benchmarks for these similar tasks. The practical value of this benchmark could be limited.

**Documentation:**

It would be great if the authors could provide a website document instead of just README.

**Ethics:**

No.

**Limitations:**

Yes

**Opportunities For Improvement:**

1. It is not very clear the difference between Meta BBO tasks and traditional hyperparameter search methods. Can the existing hyperparameter search methods also be used in this task?
2. Meta BBO seems to be also a BBO problem. Since there are already several BBO benchmarks, the value of the current benchmark could not be significant.
3. It would be great if the authors could provide a website document instead of just README.

**Relation To Prior Work:**

Yes

**Summary And Contributions:**

This paper introduces a benchmark for Meta-Black-Box Optimization with Reinforcement Learning, which aims to tune the black-box optimizers by optimizing the configurations. The key contribution of this paper is providing a template to simplify and standardize the development process. It also provides a large number of instances and baselines.

---

> ### Author Response · Authors · 2023-08-20
> **Response**
>
> Dear reviewer #t7Zi:
>
> We sincerely appreciate your thorough review and valuable feedback on our paper. We have carefully addressed each of your comments and provided a point-by-point response below.
>
> **[Difference between MetaBBO and HPO]**  While MetaBBO and HPO share similarities due to their common bi-level optimization structure, it is important to note that HPO can be considered a subset of MetaBBO. HPO focuses primarily on tuning the hyperparameters of the low-level optimizer. In contrast, the scope of MetaBBO extends beyond hyperparameter tuning, encompassing several additional dimensions of optimization. These dimensions could involve applying the meta-level controller for operator selection (e.g., [DEDDQN](https://dl.acm.org/doi/abs/10.1145/3321707.3321813)), configuring algorithms by considering both hyperparameters and operators (e.g., [RLHPSDE](https://www.sciencedirect.com/science/article/abs/pii/S2210650222001602)), and even generating optimization procedures (e.g., [Meta-GA](https://dl.acm.org/doi/abs/10.1145/3583131.3590496)).
>
> **[Difference between MetaBBO and BBO and the Significance of a MetaBBO Benchmark]** We would like to clarify that the focus of MetaBBO is on the meta-level *decision process*, which governs the behavior of the low-level optimizer. This decision process can be formulated as a *Markov Decision Process (MDP)*, allowing optimization through reinforcement learning algorithms (MetaBBO-RL). Compared with traditional BBO methods, MetaBBO methods alleviate the manual fine-tuning burden by automating the optimization of low-level BBO optimizers. They demonstrate the ability to generalize and address previously unseen problems through extensive training on a problem distribution. The development of MetaBBO benchmarks holds the potential for advancing this research direction.
>
> In MetaBox, we made serval efforts to facilitate the design and testing of new MetaBBO-RL algorithms (Section 3.1). We develop a *MetaBBO-RL coding template* that abstracts two classes: the meta-level reinforcement Agent and the lower-level Optimizer, which are implemented together with a streamlined Train-Test-Log interface, ensuring an automated workflow for design, training, and testing. Additionally, while conventional BBO benchmarks primarily assess optimization performance, MetaBox goes beyond to evaluate *generalization and knowledge transfer capabilities*. We propose two novel metrics, Meta Generalization Decay (MGD) and Meta Transfer Efficiency (MTE) to evaluate learning ability of a MetaBBO method (Section 3.4). The above new features make Metabox different from conventional BBO benchmarks, addressing the lack of a unified benchmark platform for MetaBBO methods. We also compared MetaBox and other BBO benchmarks in Table 1, underscoring the unique characteristics and novelty of our platform.
>
> **[Website Documentation]**  Following your suggestion, we have reorganized the README structure and created a separate user guide documentation, to provide clearer and more comprehensive resources for users. You can find the revised README [here](https://github.com/GMC-DRL/MetaBox) and the dedicated user guide documentation [here](https://gmc-drl.github.io/MetaBox/). Furthermore, we are committed to maintaining and updating the MetaBox platform to provide ongoing support and improvements.

---

> > ### Comment · Reviewer_t7Zi · 2023-08-30
> > **Thank you for the response**
> >
> > The authors' response addressed my questions and suggestions. I will increase my score.

---

### Official Review · Reviewer_D9Jb · 2023-07-20
**Good paper.**

**Rating:** 7
**Confidence:** 1
**Correctness:** Appears correct and detailed
**Clarity:** Well written and clear.

**Strengths:**

Exhaustive nature of experiments, the availability of problem instances and the number of problem instances available. These are a big plus.

The paper is very well written and is clearly readable.

I like the unified benchmark that is provided here.

**Additional Feedback:**

I liked the paper,  In my opinion, any benchmark should be exhaustive in its potential for diversity in problem instances and baseline methods. This is a side, this paper handles very well and therefore should be accepted.


**Documentation:**

Very detailed documentation.

**Ethics:**

None that I could find.

**Limitations:**

The limitations are fairly generic and written as an afterthought. However, this is standard practice in most papers now and therefore, not a criticism but an observation

**Opportunities For Improvement:**

First off, I am someone who has a rudimentary knowledge of black box optimization but is unfamiliar with metaBBO RL. Therefore, before getting into MetaBox section, it will be nice to explain/formulate the problem in showing which part is BBO, which part of the problem is RL and which part is estimated by the MetaBBO-RL estimation. This is attempted in section 2 but, it is too condensed and unclear to me. I suggest a separate background section with maybe a figure.

A pseudocode on usage would also benefit this paper.





**Relation To Prior Work:**

Good amount of literature review.

**Summary And Contributions:**

The paper provides a benchmark for identifying low level hyper parameters through an RL setting.  The number of available problem instances are significant, the amount of algorithms available are significant as well. As far as I could sense, the work is novel and in fact, the paper is well written

---

> ### Author Response · Authors · 2023-08-20
> **Response**
>
> Dear reviewer #D9Jb:
>
> We sincerely appreciate your thorough review and positive feedback on our paper. We have carefully addressed each of your comments and provided a point-by-point response below.
>
> **[Bi-level nature of MetaBBO]** MetaBBO aims to refine the black-box optimizers by identifying optimal configurations that boost the overall performance across various problem instances within a given problem distribution. This paradigm can be naturally defined as a bi-level optimization framework, where the meta-level enhances the performance of lower-level black-box optimizers. In the context of MetaBBO-RL, *the meta level operates as an RL agent, while the lower level encompasses a BBO optimizer*. The RL agent at the meta level estimates an action (algorithmic configuration) for the lower-level optimizer and accumulates rewards (meta performance) observed by executing the lower-level optimizer using the estimated action. Mathematically, the meta objective of the RL agent can be formulated as: $\mathop{\max}\limits_{\theta} E_{f\sim D, \pi_\theta}[ \sum_{t=0}^{T}r_t(\Lambda, f, \pi_\theta) ]$, where $D$ denotes the given problem distribution, and $r_t(\cdot)$ denotes the meta performance of the lower-level optimizer $\Lambda$ optimizing the target problem $f$, all while adhering to the algorithmic configuration provided by the meta optimizer $\pi_\theta$.  We have provided detailed explanations on this matter in the revised Section 2.
>
> **[Pseudocode for Usage]** We acknowledge the value of providing pseudocode on usage. We have taken your feedback into account and included the pseudocode for the core components in our MetaBox (such as Trainer and Tester) in Appendix F, along with corresponding descriptions.

---

> > ### Comment · Reviewer_D9Jb · 2023-08-29
> > **Thank you for your response**
> >
> > I am generally satisfied with this paper. Thank you for your effort in the responses.

---

### Official Review · Reviewer_qtiV · 2023-07-21

**Rating:** 8
**Confidence:** 4
**Clarity:** Yes, the paper is well written.

**Strengths:**

- The writing is clear; the authors comprehensively describe the benchmarking problems and how the benchmark is composed, both in terms of concepts and code implementation.
- The benchmark suite is well-designed in terms of code implementation; each module is abstracted effectively, enabling users to focus solely on algorithm implementation without needing to understand the benchmark suite's internals.
- MetaBox offers a wide range of test suites, including both synthetic and realistic scenarios.

**Additional Feedback:**

The main evaluation setting for the number of function evaluations (maxFE) is perceived as quite unrealistic, particularly for black-box optimization involving high evaluation costs, as mentioned in lines 19-20. To address this concern, it would be beneficial to include the performance of the baseline methods over the number of function evaluations. This addition will offer a clearer perspective and enable a better understanding of how the baseline methods behave under different evaluation settings.

By incorporating this information, readers can gain insights into the effectiveness and efficiency of the baseline algorithms as the number of function evaluations varies, providing a more comprehensive assessment of their performance.

**Correctness:**

Most of the results are well-measured and appear legitimate. However, there are a few areas where the presentation could be improved.

The bar charts lack information about the variations observed in the experiments. For example, in Figure 3, there is no representation of the performance variations across the runs. It would be beneficial to include error bars or confidence intervals to show the extent of variation exhibited by each baseline algorithm.

By adding these details, readers would gain a better understanding of the consistency and reliability of the baseline performances. This enhancement would provide a more comprehensive and transparent evaluation of the proposed benchmark platform and algorithms.

**Documentation:**

The authors have provided the code and its appropriate documentation through the shared link. Additionally, in the discussion section of the paper, they offer a brief explanation of the maintenance plan and outline their future implementation plans.

**Limitations:**

The authors acknowledge a potential limitation in their paper, specifically regarding the performance evaluation of BBO. They recognize that this aspect remains an open question, meaning that there might not be a definitive or universally accepted method to assess the performance of Black-Box Optimization algorithms.

However, they do not specify further details or elaborate on this limitation in the given text. Depending on the context of the paper and the section where this statement is made, it might be useful for the authors to explain why the performance evaluation of BBO is challenging or open-ended. They could also discuss existing approaches or research gaps in this area. Providing more context and insights about this limitation can help readers understand the scope and implications of their work better.

**Opportunities For Improvement:**

While most components in the paper are self-explanatory, further elaboration on the type of benchmarking algorithms would greatly enhance readability and comprehensibility for readers. As mentioned in lines 32-33, the target benchmarking settings are formulated within a bi-level optimization framework, where the upper level involves optimizing the meta-level optimizers, and the lower level pertains to the optimization routines of black-box optimizers. Although the current writing briefly explains these targets and frameworks in lines 35-37, a more detailed discussion is warranted to provide readers with a deeper understanding of the methodology employed.

**Relation To Prior Work:**

Yes, the authors explain the relationship to prior works while highlighting the distinctive features of MetaBox compared to other works.

**Summary And Contributions:**

This paper proposes MetaBox, a benchmark platform for Meta-Black-Box Optimization (MetaBBO) using Reinforcement Learning (RL). MetaBox offers a unified yet flexible algorithmic template, allowing users to effortlessly interface and evaluate new BBO algorithms in a controlled and standardized manner. The platform includes over 300 BBO problem instances collected from synthetic and realistic scenarios, along with 19 baseline (Meta) BBO methods. Additionally, MetaBox introduces three standardized performance metrics designed to assess the baseline algorithms consistently. Lastly, the authors present the benchmarked results of BBO algorithms conducted using the proposed MetaBox.

---

> ### Author Response · Authors · 2023-08-20
> **Response**
>
> Dear reviewer #qtiV:
>
> We sincerely appreciate your thorough review and positive feedback on our paper. We have carefully addressed each of your comments and provided a point-by-point response below.
>
> **[Detailed Explanation of Benchmarking Algorithms]** The benchmarking algorithms included in this study fall within the realm of MetaBBO, characterized by their bi-level optimization nature. Specifically, MetaBBO aims to refine the black-box optimizers by identifying optimal configurations that boost the overall performance across various problem instances within a given problem distribution. This paradigm can be naturally defined as a bi-level optimization framework, where the meta-level enhances the performance of lower-level black-box optimizers.  At the meta level, the meta optimizer (often parameterized by neural networks, e.g., $\pi_\theta$) tailors the configuration of the lower-level optimizer based on the optimization status of the latter at the current time step. Then, the meta optimizer observes the next-step optimization performance (meta performance) of the lower-level optimizer, and trains itself by maximizing the accumulated meta performance. At the lower level, once equipped with the algorithmic configuration, the optimizer carries out optimization for the target task, observes the objective value change between steps, and communicates this change back to the meta optimizer as the meta performance. As per your suggestion, we have incorporated this extended definition at the start of Section 2, along with the inclusion of a conceptual figure (Figure 2).
>
> **[Clarification of Performance Evaluation Limitation]** In MetaBox, we addressed the performance evaluation challenge by implementing three standardized metrics aimed at evaluating both optimization performance and the learning effectiveness of MetaBBO-RL approaches. Notably, we introduce a novel Aggregated Evaluation Indicator (AEI) that offers a holistic view of BBO optimization performance. The AEI scoring system takes into consideration three main aspects: the best objective value achieved, the convergence rate, and the algorithmic complexity. This comprehensive approach covers aspects that have been extensively discussed in the literature. Nevertheless, *it is important to acknowledge that the evaluation of BBO performance is not a one-size-fits-all endeavor.* Different practical applications may have varying preferences and additional concerns. For example, in certain domains, factors such as *solution robustness, solution diversity, parallelization and scalability*, might play a crucial role in assessing the true efficacy of an optimizer. These nuances extend beyond the scope of a single evaluation metric. This is why, in our conclusion section, we stated that the performance evaluation of BBO remains a challenging and open-ended area. In response to your suggestion, we have taken this opportunity to provide additional context and insights about this limitation in our revised paper (Section 5, page 10).
>
> **[Providing Variation Information in Bar Charts]** Thank you for the suggestion. Following it, we have incorporated error bars for each baseline algorithm into the AEI scores bar charts **(Figure 4, page 8; also Figure 1, Appendix, page 4)**. The error range associated with each baseline algorithm is calculated as the standard deviations of its AEI scores $[AEI_k]_{k=1}^{K}$ across the *K* tasks in the testsuites. Algorithm with a smaller error bar performs more stably on the target testsuites.
>
> **[Performance over the Increment of FEs]** We strongly agree with your concern regarding the significance of managing the number of function evaluations (FEs), particularly in high-evaluation-cost scenarios. In MetaBox, the default FEs for Synthetic testsuites is set to $2 \times 10^4$ (line 19-20 in Appendix). This choice adheres to the standard practice in benchmarking BBO optimizers on Synthetic testsuites. *For the realistic Protein-Docking testsuites in MetaBox, due to its computationally expensive nature, the default FEs is set to $10^3$* (see line 38-39 in Appendix).  Besides, it is important to note that MetaBox offers the flexibility of accommodating user-specific FEs. Users can easily tailor the evaluation process by adjusting the maxFEs parameter within the training or evaluating interface.
>
> We also appreciate the suggestion of presenting performance over the increment of FEs. We wish to clarify that our platform already incorporates the capability to automatically generate cost curves that showcase the performance of different algorithms relative to varying FEs. But due to the page limit, this information is illustrated in the Appendix D.1, Figure 2, Cost Curve part and the complete results can be accessed through our online webpage [here](https://anonymous.4open.science/r/MetaBox-6F0C/post_processed_data/content.md).

---

> > ### Comment · Reviewer_qtiV · 2023-08-28
> >
> > Thanks for the updates!
> >
> > I'm actually quite impressed by the results that the author shared. I'm cranking my evaluation up a notch. Looking forward to more insightful work from you in the future!

---

> > > ### Author Response · Authors · 2023-08-28
> > >
> > > We sincerely appreciate the reviewer for acknowledging our work and cranking his/her evaluation up. We will continuously contribute to the BBO community in the future!

---

### Official Review · Reviewer_nMBF · 2023-07-21
**Valid and well described benchmark platform for Meta-Black-Box Optimization with Reinforcement Learning**

**Rating:** 9
**Confidence:** 4

**Strengths:**

Comprehensive platform with both a large number of problem instance, a significant number of state of the art optimization algorithms and standardized performance metrics.

This permits reproducible research in the MetaBBO-RL field.

The three main contributions constituting the MetaBBO-RL platform, i.e. MetaBBO-RL template approach, a large set of benchmark instances and a representative set of baseline optimization algorithms are clearly introduced and presented in detail in dedicated subsections in section 3.

A comprehensive benchmarking study has been conducted and even if it is not the main purpose of the work, some results did outperform the state-of-the-art, which outlines the potential of the proposed MetaBBO-RL approach.

**Additional Feedback:**

Interesting contributions for NeurIPS and for the RL for optimisation community.

**Clarity:**

The article is very well structured and written.

The basic notions of blackbox optimisation and meta-black box optimisation are clearly introduced.

**Correctness:**

Claims are properly backed up with references.

Experiments are conducted with scientific rigour.

**Documentation:**

A comprehensive documentation is provided on the Github repository. The latter includes a quick start guide, and different “how-to”.

Section 4 also provides a brief overview of the different steps to use the MetaBox platform.


**Ethics:**

No ethical concerns raised as only classical optimization benchmarks are involved.

**Limitations:**

Some limitations have been identified in section 5: lack of performance evaluation for BBO, expanding the number of BBO tasks in the platform and proper maintenance of the platform, keeping it up-to-date with latest algorithms from the literature.

**Opportunities For Improvement:**

The definition of bi-level optimization could be (even briefly) introduced.

**Relation To Prior Work:**

The authors refer to the (limited) previous work in the field of MetaBBO. These are relevant and recent ones.

The novelty of the proposed platform is clearly outlined, i.e. the lack of a unified benchmark platform for MetaBBO-RL.

Existing BBO benchmarks are listed and compared to the proposed MetaBox.

The MetaBox platform also takes advantages of existing libraries like for some optimization algorithms thanks to API calls to DEAP and Scikit-Optimizer.

**Summary And Contributions:**

This article introduces MetaBox, a novel benchmark platform for meta-black-box optimization using reinforcement learning.

The platform comes with a large set of meta-black-box problem instances (more than 300) and a set of 19 baseline optimizers.

---

> ### Author Response · Authors · 2023-08-20
> **Response to Reviewer #nMBF**
>
> Dear reviewer **#nMBF**:
>
> We sincerely appreciate your thorough review and positive feedback on our paper. We have carefully addressed each of your comments and provided a point-by-point response below. We have also uploadedd the revisied version of paper to the system.
>
> **[Definition of Bi-level Optimization in MetaBBO]** MetaBBO aims to refine the black-box optimizers by identifying optimal configurations that boost the overall performance across various problem instances within a given problem distribution. This paradigm can be naturally defined as a bi-level optimization framework, where the meta-level enhances the performance of lower-level black-box optimizers.  Specifically, at the meta level, the meta optimizer (often parameterized by neural networks, e.g., $\pi_\theta$) tailors the configuration of the lower-level optimizer based on the optimization status of the latter at the current time step. Then, the meta optimizer observes the next-step optimization performance (meta performance) of the lower-level optimizer, and trains itself by maximizing the accumulated meta performance. At the lower level, once equipped with the algorithmic configuration, the optimizer carries out optimization for the target task, observes the objective value change between steps, and communicates this change back to the meta optimizer as the meta performance. Mathematically,  the bi-level optimization objective can be formulated as: $\mathop{\max}\limits_{\theta} E_{f\sim D,  \pi_\theta}[ \sum_{t=0}^{T}r_t(\Lambda, f, \pi_\theta) ]$, where $D$ denotes the given problem distribution, and $r_t(\cdot)$ denotes the meta performance of the lower-level optimizer $\Lambda$ optimizing the target problem $f$, all while adhering to the algorithmic configuration provided by the meta optimizer $\pi_\theta$. As per your suggestion, we have incorporated this extended definition at the start of Section 2, along with the inclusion of a conceptual figure (Section 2, page 3).
>
> **[Limitations Discussed in Section 5]** Yes, as a pioneering benchmark platform in the MetaBBO field, we acknowledged there are still certain limitations of our MetaBox and outlined them in Section 5. We are fully committed to addressing these limitations by maintaining the platform and regularly updating its Baseline Library, test suites, performance evaluation metrics, and other integral components. We also welcome user feedback to assist us in enhancing the quality and scope of our platform.

---

> > ### Comment · Reviewer_nMBF · 2023-08-30
> >
> > Thank you for the clarifications and for updating your article.
> >
> > I obviously keep the ranking I initially assigned.

---

### Author Response · Authors · 2023-08-20
**Global Response**

We would like to express our sincere gratitude for the time and effort the reviewers have invested in reviewing our paper. We are also pleased to see the reviewers have recognized our MetaBox platform of being **comprehensive** (all 4 reviewers), **well-written** (all 4 reviewers), **reproducible** (#nMBF), **well-designed** (#qtiV), **unified** (#D9Jb) and **novel** (#nMBF). In this global response, we primarily address a common suggestion shared by the reviewers and provide an overview of the modifications suggested by each reviewer, as follows.

---

**[More elaboration on the Bi-Level Nature of MetaBBO Paradigm]** We thank the reviewers (#nMBF, #qtiV, and #D9Jb) for raising this valuable point, and we apologize for any prior confusion. Following the suggestion, we have included a dedicated paragraph at the beginning of Section 2 (page 3), along with a newly introduced conceptual figure (Figure 2), to provide a clearer illustration of the bi-level nature of the MetaBBO paradigm. Here is the additional elaboration we have incorporated:

*MetaBBO methods operate within a bi-level optimization framework designed to automate the fine-tuning process for a given BBO optimizer. Distinguishing themselves from conventional BBO techniques, MetaBBO methods introduce a novel meta-level as an automatic decision process. The purpose is to alleviate the need for labor-intensive manual fine-tuning of lower-level BBO optimizers. Typically, they require the ability to generalize behaviors to address previously unseen problems through extensive training on a problem distribution.*

- At *the meta level, the meta optimizer (e.g., an RL agent) dynamically configures the lower-level optimizer based on the current optimization status at that particular time step. Then, the meta optimizer evaluates the performance of the low-level optimizer over the subsequent optimization steps, referred to as meta performance. The meta optimizer leverages this observed meta performance to refine its decision-making process, training itself through the maximization of accumulated meta performance, thereby advancing its meta objective.*
- At *the lower level, the BBO optimizer receives a designated algorithmic configuration from the meta optimizer. With this configuration in hand, the low-level optimizer embarks on the task of optimizing the target objective. It observes the changes in the objective value across consecutive steps and transmits this information back to the meta optimizer, thereby contributing to the meta performance signal.*

**[Other modifications]** We add a paragraph to discuss the remained open-ended question (***#qtiV***) about performance evaluation in BBO (**Section 5, page 10**). Besides, we add error bars to AEI scores bar charts (**Figure 4, page 8; also Figure 1, Appendix, page 4**) to reflect robustness of a baseline algorithm across different problems in a problem distribution (***#qtiV***). We also add detailed pseudo codes (**Appendix F**) aiding for elaborating the complete benchmarking process in MetaBox (***#D9Jb***).

---

### Decision · Program_Chairs · 2023-09-22

**Decision:**

Accept (Oral)

**Comment:**

This paper investigates benchmark datasets and methods for meta-black-box optimization with reinforcement learning. The reviewers and I agree that the platform is extensive and covers a large set of relevant problem instances and set of algorithms for solving them. And, the code to solve these is well-designed and will serve as a useful tool for the community. This paper will be a great contribution to this year's program. I recommend it for an oral as these benchmarks and methods are well-executed and worth emphasizing to the community.